# Beyond Solving:
# A Closer Look at LLMs as Solution Verifiers

## Abstract

Large language models (LLMs) can act as both problem solvers and solution verifiers, with verifiers improving solver performance by selecting high-quality answers from a pool of candidates. However, prior studies of solver–verifier interactions have been limited, focusing mainly on self-verification and rarely examining how verifiers judge outputs from models in their own or in another model family. Modern LLMs also undergo extensive post-training, but its effect on verification remains unclear. We present a systematic study across 37 models spanning multiple families, sizes, and base vs. post-trained variants, evaluated on 9 benchmarks covering logical reasoning, structured puzzles, symbolic computation, mathematics, commonsense, factual recall, and domain knowledge. We compare self-verification with verification within the same family and across different families. To support this, we introduce and empirically validate *verifier gain*, a metric that predicts the performance improvements from *test-time verifier-based rejection sampling*. We analyze how metrics like verifier gain and false positive rate scale with model size and post-training, and characterize differences in dataset verifiability. Our findings show that cross-family verification is especially effective; post-training reduces self-improvement but strengthens cross-family improvement; and mathematical and logical tasks exhibit the highest inherent verifiability.

## 1 Introduction

Problem-solving with LLMs has shifted from solely querying a model for a solution to a system where models both solve and verify. The paradigm of verifying solutions at test time is broad, spanning both simple strategies, such as generating multiple candidate solutions and using a verifier as a filter (Zhao et al., 2025), and more complex iterative refinement approaches (Madaan et al., 2023). With generative verification at test time, LLMs can solve more complex problems than they can when used alone (Cobbe et al., 2021; Lightman et al., 2024).

Despite the increasing dominance of this paradigm, studies of solver–verifier interactions have remained limited in scope. Prior work has largely examined how a single model verifies its own solutions (self-verification) and improves itself (self-improvement) (Song et al., 2025), yet self-verification is not guaranteed to be effective: models may be biased toward their own reasoning patterns, their training may reinforce these tendencies, and different tasks may vary in how much they benefit from verification. Additionally, this focus on self-verification offers little insight into how verification behaves when the solver and verifier differ. With open-source model families that have base and post-trained pairs, size variants, reproducible inference pipelines, and explicit reasoning traces, we can systematically study verification across models. We therefore broaden our analysis to include both intra-family and cross-family verification and ask the following central question:

> *When does verification actually pay off, and how does each factor, such as model family, model size, post-training, solver–verifier similarity, or task type, influence how effective verification is in improving the solver?*

To accomplish this, we evaluate verifiers across a diverse suite of tasks, including synthetic tasks used to test precise logical reasoning or symbolic computation (3-SAT, Sudoku, and matrix multiplication), mathematical reasoning tasks (AIME (Mathematical Association of America, 2025), GSM8K (Cobbe et al., 2021)), commonsense and factual reasoning (CSQA (Talmor et al., 2019), GPQA (Rein et al., 2024)), and broad domain knowledge (MMLU in STEM and social sciences (Hendrycks et al., 2021)) using 37 models from 7 model families.

Our analysis indicates that self-verification does not always "pay off": models often favor solutions resembling their own reasoning (Section 5.1, 5.3), post-training can sharpen this bias (Section 5.4), and some tasks inherently benefit less from verification than others (Section 5.5). Therefore, we present the following contributions, which offer actionable and empirically supported guidance for how to use verifiers effectively.

- **New Metric: Verifier Gain.** Verifier accuracy alone provides an incomplete picture of verifier usefulness at test time. To address this, we derive *verifier gain*, a metric that simulates the improvement obtained from a verifier during test-time rejection sampling. We empirically study rejection sampling with verifiers and show that this theoretical formulation closely reflects empirical performance trends.

- **Self-Improvement, Intra-Family Improvement, and Cross-Family Improvement.** We extensively compare performance improvements from self-verification, intra-family verification, and cross-family verification, finding that cross-family verification is often the most beneficial, particularly when compared to self-verification. We link these differences to similarities in the solution distributions of the solver and verifier: verifier gain decreases as these distributions become more similar. Our results suggest that as models become stronger, whether through increased scale, post-training, or simply higher solver accuracy, they become less effective as self-verifiers and more effective as cross-family verifiers.

- **Dataset Verifiability.** We study whether tasks that are easy to solve are also easy to verify and whether some tasks are inherently more verifiable than others. We find that verification accuracy generally correlates with solver accuracy, though self-verification yields little verifier gain across all tasks. We also observe that a clear subset of tasks involving mathematical or logical reasoning consistently produces higher verifier gains.

## 2 RELATED WORK

**Verifiers.** Broadly, verifiers can operate on the *outcome* level, by judging only the correctness of the final answer, or the *process* level, by judging the correctness of the intermediate reasoning steps. Weng et al. (2023), Wu et al. (2024), and Jiang et al. (2024) develop methods for outcome-level self-verification by predicting parts of the question conditioned on the solution. In order to reduce hallucinations, Dhuliawala et al. (2024) have language models fact-check their own generations by generating fact-check questions. Researchers have also trained general-purpose outcome level verifiers (Hosseini et al., 2024; Zhang et al., 2025; Cobbe et al., 2021) and value models (Yu et al., 2024), either independently or jointly with the solver (Shen et al., 2021; Sareen et al., 2025). Work on process-level verification has focused on deductively verifying Chain-of-Thought (CoT) reasoning (Ling et al., 2023), verifying individual proof steps (Yang et al., 2022), and training process reward models for mathematical reasoning (Luo et al., 2025). Finally, Song et al. (2025) investigate the performance improvement caused by using an outcome-level verifier (the GV-Gap), and how this improvement changes as the solver or verifier increases in capacity. However, they primarily focus on cases where the solver and verifier are the same model. Additionally, they only study base models and do not consider post-trained models in their analysis.

**Scaling test-time compute.** Recently, prior work focused on studying scaling test-time compute with verifiers. For example, Zhao et al. (2025) study random sampling with self-verification, Chen et al. (2025) study combining parallel sampling with self-correction, and Singhi et al. (2025) compare Self-Consistency (Wang et al., 2023) to scaling with a generative verifier. Snell et al. (2025) investigate compute-optimal approaches to test-time scaling with process-level verification. Finally, verifiers also have their limitations. Imperfect verifiers can produce false positives (Stroebl et al., 2024), eliminate valid reasoning paths (Yu et al., 2025), and fail to select the right solution (Brown et al., 2024). We present additional related work on test-time verification in Appendix A.

**Self-improvement.** Researchers have also studied LLM self-improvement and self-evaluation, with some voicing skepticism (Huang et al., 2024b; Kamoi et al., 2024; Olausson et al., 2023; Panickssery et al., 2024). On the other hand, recent work has provided a theoretical framework for self-improvement via distribution sharpening, and empirical support alongside (Huang et al., 2024a). Zhang et al. (2024) look specifically at self-improvement for small models, arguing that they need to be paired with a stronger verifier. Some practical methods for self-improvement use natural language feedback (Madaan et al., 2023; Shinn et al., 2023; Kim et al., 2023) or train models

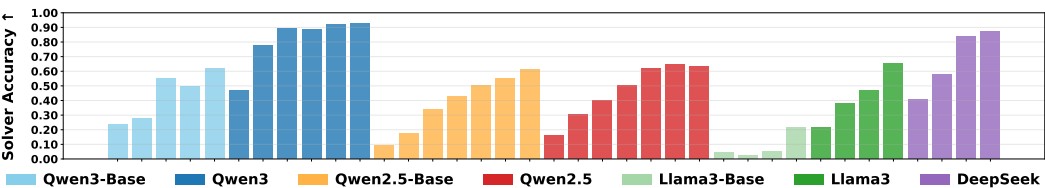

Figure 1: Average solver accuracy of each model over all datasets. Base model families are suffixed by **-Base**. Models within each family are ordered in increasing sizes.

for self-correction explicitly (Welleck et al., 2023). Other methods use tools (Gou et al., 2024), particularly code interpreters (Zhou et al., 2024), to iteratively improve solutions.

## 3 PRELIMINARIES

In this section, we establish the framework used throughout this work. We define datasets, solvers, and verifiers, introduce the metrics used to evaluate solver and verifier behaviors, and specify the verification settings in our empirical analysis.

### 3.1 DATASET, SOLVERS, AND VERIFIERS

Let $\mathcal{D} \subseteq \mathcal{X} \times \mathcal{Y}^\star$ be a dataset of pairs $(x, \mathcal{Y}_x)$, where $x \in \mathcal{X}$ is a problem and $\mathcal{Y}_x \subseteq \mathcal{Y}$ is a non-empty set of correct solutions. A solver $S : \mathcal{X} \to \mathcal{Y}$ is an LLM that produces a solution $y$ for a given problem $x$, and a verifier $V : \mathcal{X} \times \mathcal{Y} \to \{0, 1\}$ is an LLM that evaluates a problem–solution pair and returns a binary judgment. Following Song et al. (2025), who find chain-of-thought (CoT) verification more stable than multiple-choice formats, we instruct both solvers and verifiers to generate CoT reasoning before producing their final solutions and judgments. We define the correctness indicator as $c(x, y) = \mathbb{1}\{y \in \mathcal{Y}_x\}$.

### 3.2 EVALUATION METRICS

We define the accuracy of a solver $S$ on a dataset $\mathcal{D}$ as the expected correctness of its outputs over all problems in the dataset: $\mathbb{E}_{(x, \mathcal{Y}_x) \sim \mathcal{D},\, y \sim S(x)} \big[ c(x, y) \big]$. Verifier performance has several dimensions. We report common binary classification metrics, including verifier accuracy, false positive rate (FPR), false negative rate (FNR), F1-Score, and precision, with their definitions in Appendix B.

Our primary goal is to evaluate whether using a verifier $V$ can improve a solver $S$ at test time via rejection sampling, in which solver outputs are repeatedly sampled until the verifier accepts one. Assuming the solver has a non-zero probability of sampling a correct solution, and in the limit of infinite resampling, the expected correctness of the accepted solution converges to the verifier's precision, i.e., the proportion of accepted solutions that are actually correct. To quantify the improvement from combining a solver with a verifier, we define **verifier gain**:

$$\text{Gain}(S, V; \mathcal{D}) = \text{Precision}(S, V; \mathcal{D}) - \text{SolverAcc}(S; \mathcal{D}). \tag{1}$$

It is worth noting that verifier gain is an asymptotic metric: it reflects the limit of infinite sampling and therefore serves as a bound on the improvement attainable by verifier-based rejection sampling. Throughout this work, we use verifier gain to compare differences in verifier behavior.

### 3.3 VERIFICATION SETTINGS

We group models into families (e.g., `Llama3`, `Qwen2.5`), where each family contains related models of varying sizes. Because base and post-trained models often exhibit substantially different behaviors, we treat them as distinct families. For example, the base model `meta-llama/Meta-Llama-3-70B` belongs to the `Llama3-Base` family and the post-trained model `meta-llama/Meta-Llama-3-8B-Instruct` belongs to the `Llama3` family.

We categorize each solver–verifier pair from our pool of models into one of three verification settings. Each solver–verifier pair is considered an instance of:

1. **Self-Verification.** The solver and verifier are the same model, so the model verifies its own solutions. For example, when a 70B `Llama3` model is used as both the solver and verifier, the verification metric (e.g., accuracy, FPR) is computed on this single pairing.

2. **Intra-Family Verification.** The verifier evaluates solutions produced by other models within the same family. For example, a 70B `Llama3` verifier may evaluate outputs from 8B or 13B

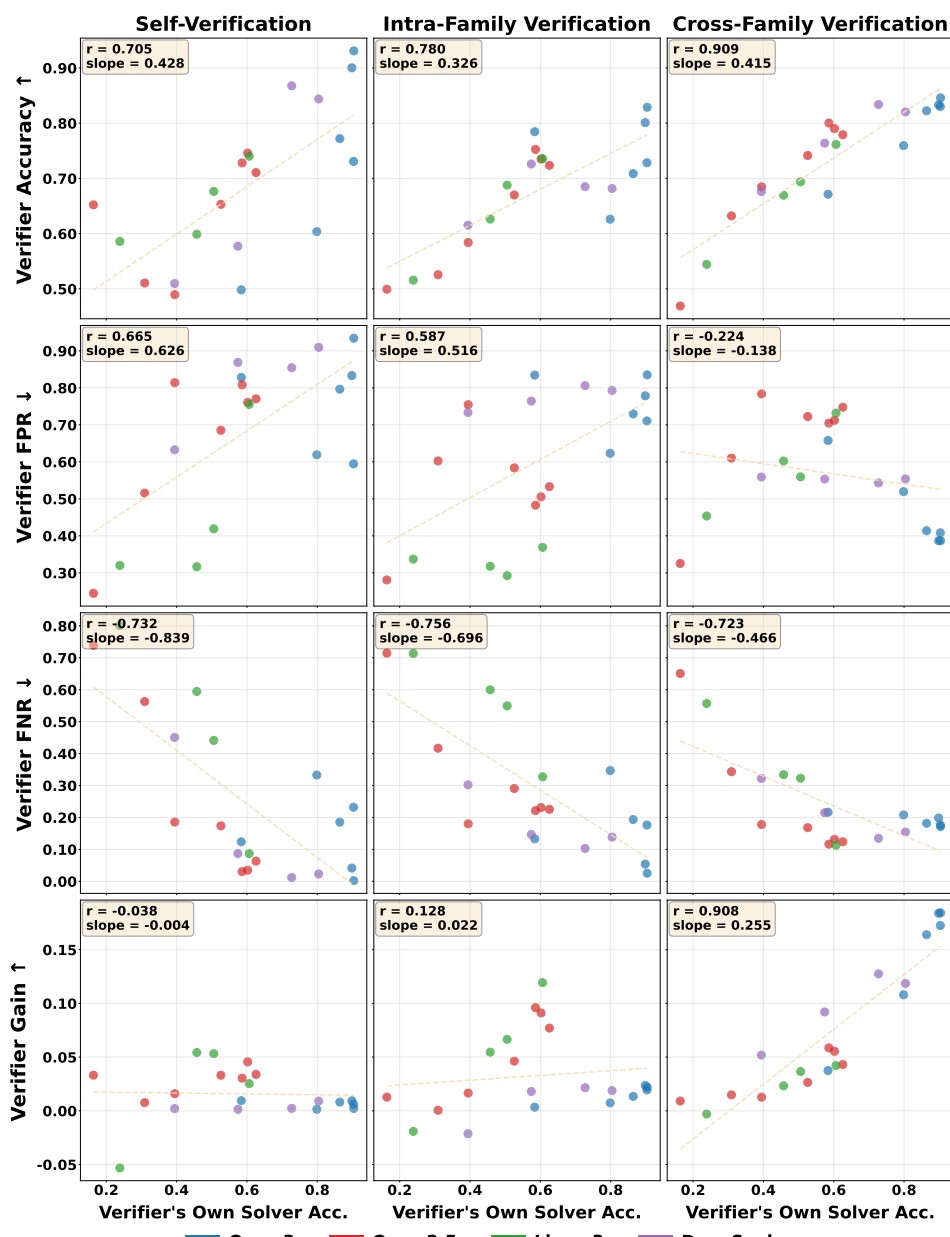

Figure 2: Correlation between each verifier's metrics (rows) and its own solver accuracy for all 21 post-trained models, averaged over all datasets. Each verifier metric is computed over our three verification settings (columns).

`Llama3` solvers. The reported metric is averaged over all such within-family solvers, excluding the self-verification case.

3. **Cross-Family Verification.** The verifier evaluates solutions produced by models from different families. For example, a base `Llama3` verifier may evaluate outputs from `Qwen3` or from a post-trained `Llama3`. The reported metric is averaged over all such cross-family solvers.

Using these three categorizations, we evaluate a verifier by applying it to a set of solver models, computing the corresponding verifier metrics, and then partitioning these metrics according to the three verification settings and averaging within each partition. A formal mathematical description of each verification setting is provided in Appendix C.

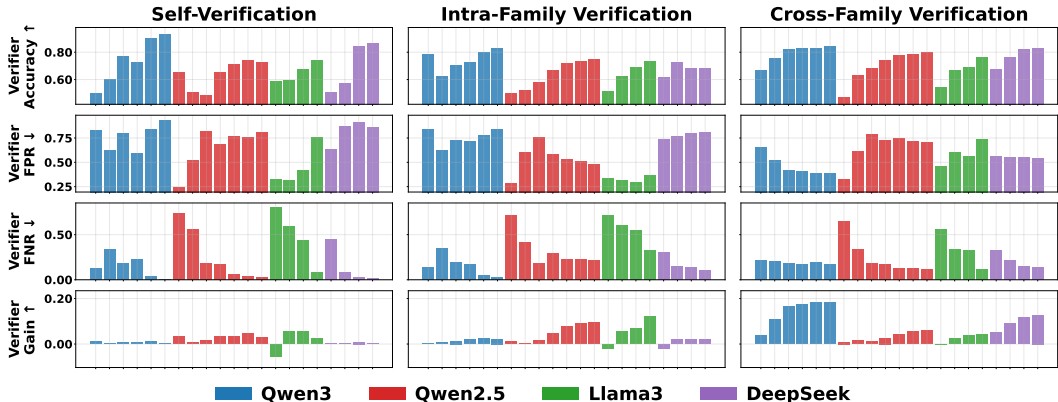

Figure 3: Correlation between each verifier's metrics (rows) and model size for all 21 post-trained models, averaged over all datasets. In each plot, models are separated by family and ordered by increasing size. Each verifier metric is computed over our three verification settings (columns).

## 4 EXPERIMENTAL SETUP

**Models.** We evaluate the solver and verifier abilities of 21 post-trained models from the `Llama3` (Grattafiori et al., 2024), `Qwen2.5` (Qwen et al., 2024), `Qwen3` (Yang et al., 2025), and `DeepSeek-R1` (DeepSeek-AI et al., 2025) families. For our study of post-training effects in Section 5.4, we additionally evaluate 16 base models from the `Llama3-Base`, `Qwen2.5-Base`, and `Qwen3-Base` families. Model sizes range from 0.5B to 72B parameters. The full model list, with sizes, families, and HuggingFace identifiers, is provided in Appendix D. Figure 1's legend displays the seven model families and the color scheme assigned to each.

**Datasets.** To comprehensively evaluate each model's performance as both a solver and a verifier, we compile a broad suite of real-world and synthetic tasks spanning diverse domains. We include tasks requiring mathematical reasoning (GSM8K, AIME), commonsense knowledge (CSQA), and domain-specific factual knowledge of varying breadth (MMLU STEM, MMLU Social Sciences, GPQA). We also construct synthetic tasks to assess logical reasoning (3SAT), structured puzzle solving (Sudoku), and symbolic computation (Matrix Multiplication). Further dataset details, along with examples of our synthetic tasks, are provided in Appendix E. Code for all experiments, including synthetic-data generation, is included in the Supplementary Materials and will be open-sourced upon publication.

**Evaluation.** Datasets such as Matrix Multiplication and the natural-language benchmarks contain a single ground-truth answer per problem. For these, we extract boxed solver outputs and evaluate them via exact matching. In contrast, datasets like Sudoku and 3SAT may admit multiple valid solutions, so we evaluate solver outputs according to the rules of the respective task. To evaluate verifiers, we prompt each model to generate CoT reasoning from the problem and solver answer before producing a boxed "correct" or "incorrect", from which we extract the final judgment.

**Implementation.** For both solvers and verifiers, we generate with temperature 0.7, top-p 0.9, and a maximum output length of 8192 tokens. We discard outputs that do not contain a boxed answer. All inference experiments are run using vLLM on H200 GPUs. Prompts and additional details on output filtering are provided in Appendix F.

## 5 RESULTS

### 5.1 DO BETTER SOLVERS MAKE BETTER VERIFIERS?

**Solver performance.** We first benchmark the performance of all 37 models on each of our 9 datasets, averaging performance across tasks (Figure 1) and reporting task-level results in Appendix G. Overall, solver accuracy increases with model capacity. Models in the `Qwen3` and `DeepSeek` families perform particularly well, whereas `Llama3-Base` performs poorly due to base models being unfamiliar with the question–answering instruction format. Within each family, we observe clear performance scaling for `Qwen2.5-Base` and `DeepSeek`, with the remaining families showing similar upward trends.

**Correlating verifier and solver performance.** After establishing solver accuracy, we analyze whether a model's solver performance correlates with its performance as a verifier (Figure 2). For each of our 21 post-trained models and each dataset, we evaluate verification on the same set of solver models to obtain verifier accuracy, F1-score, precision, FPR, FNR, and gain for every solver–verifier pair. For each verifier, we then divide the verifier metrics into three verification settings and average within each setting over solvers and datasets.

Verifier accuracy tends to improve with the verifier's own solver accuracy, but the relationship becomes more nuanced when examining other metrics. The FPR increases during self-verification and intra-family verification but decreases slightly during cross-family verification. This indicates that verifiers with stronger solver abilities are more likely to incorrectly label solutions as correct when verifying their own outputs or outputs from models within their family. We provide additional visualizations for F1-score and precision in Appendix H.

To better interpret the trends suggested by the accuracy and FPR results, we examine verifier gain in the final row. Verifier gain quantifies the expected benefit of using the verifier during rejection sampling, i.e., repeatedly sampling until the verifier accepts a solution (a common solver–verifier interaction setting (Song et al., 2025)). This visualization offers a clearer view of verification quality: self-verification yields the smallest gains, and more accurate solvers do not exhibit greater self-improvement. Gains increase slightly in intra-family verification, while cross-family verification provides the greatest potential benefits.

**Examining verifier performance at different model families and sizes.** In Figure 3, we repeat the experiments from Figure 2 but plot each verifier metric as a function of model size within each model family. We observe that verification accuracy and FNR consistently improve as models become larger, whereas FPR behaves more inconsistently, often increasing with model size (e.g., intra-family verification for `DeepSeek`).

For state-of-the-art post-trained models such as `Qwen3` and `DeepSeek`, we find that verifier gains are largest in the cross-family setting, smaller in the intra-family setting, and minimal during self-verification. At first glance, this appears to contradict Song et al. (2025), who report that self-verification GV-Gaps increase with more pretraining FLOPs. However, their analysis focuses on older model families, and Figure 2 likewise shows larger verifier gains for older models such as `Qwen2.5` and `Llama3`.

We hypothesize that stronger post-trained models like `DeepSeek` and `Qwen3` show negligible gains in self-verification and limited gains in intra-family verification for two reasons: (a) they may already engage in *spontaneous* self-verification when used as solvers, reducing the benefit of an additional *forced* verification round, and (b) their distributions are significantly sharpened by post-training (Huang et al., 2024a), which limits the improvement obtained from rejection sampling.

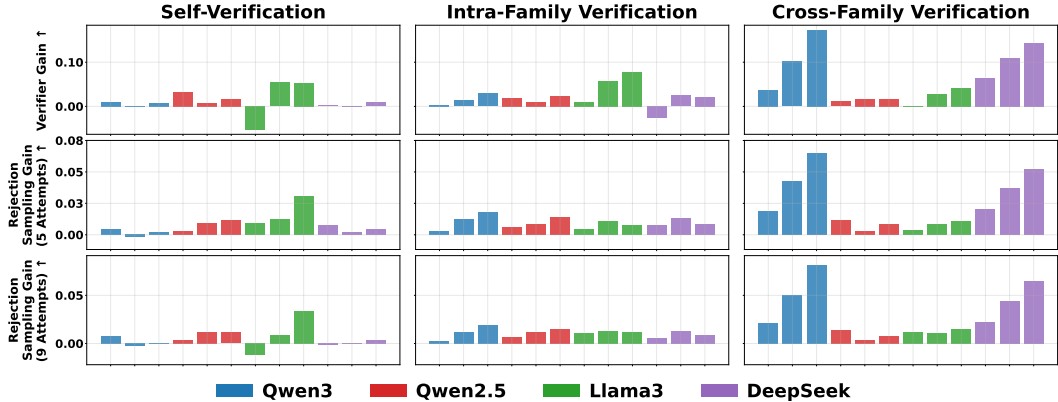

Figure 4: Comparison between theoretical and empirical verifier gains (rows) for each verification setting (columns). Row 1 shows verifier gains computed from Equation 1. Rows 2 and 3 each show the gains from rejection sampling, computed from rejection sampling using verifiers for up to 5 and 9 solver attempts, respectively.

**Takeaways**

- Verifier models are biased toward accepting incorrect solutions when performing self-verification or intra-family verification.

- Verification accuracy alone is not a reliable predictor of how much a verifier can improve a solver at test time. Instead, computing verifier gain using solver accuracy and verifier precision provides a more reliable metric.

- While model families like `Llama3` and `Qwen2.5` show some ability to self-improve based on their verifier gains, stronger model families like `DeepSeek` and `Qwen3` do not.

## 5.2 IS VERIFIER GAIN A GOOD PREDICTOR FOR IMPROVEMENTS FROM RESAMPLING?

Our verifier gain metric estimates the expected improvement in a solver's accuracy when using a verifier for rejection sampling. To assess how well this metric predicts real performance, we conduct rejection sampling experiments across all solver–verifier pairs from a 12-model subset of our post-trained models, consisting of the three smallest models from each of the four post-training families. For each problem in each dataset, the solver generates solutions until the verifier labels one as correct, for up to nine attempts; if no such solution is found, we retain the final attempt. The empirical results, along with the corresponding theoretical verifier gains, are shown in Figure 4.

Although the measured improvements can be noisy when verifier gains are small and the nine-attempt cap limits the rejection sampling procedure, the overall trends align closely with theoretical predictions. In particular, we observe smaller gains from self-verification than from both intra-family and cross-family verification, as well as stronger scaling behavior for cross-family verification relative to the other settings.

**Takeaways**

The verifier gain given by Equation 1 is a reliable predictor of performance improvements under rejection sampling. Crucially, it can be estimated from one round of verification without requiring computationally expensive rejection sampling experiments.

## 5.3 ARE VERIFIERS BIASED TOWARD SOLUTIONS THAT RESEMBLE THEIR OWN?

Humans tend to judge solutions that resemble their own reasoning as more likely to be correct. This mirrors the well-documented self-enhancement bias (Krueger, 1998), in which individuals evaluate themselves more favorably than objective evidence would suggest. Our results suggest that an analogous effect may arise in solver–verifier interactions. As shown earlier in Figures 2 and 3, strong reasoning models benefit the least from self-verification and the most from cross-family verification, hinting at a similar bias.

To directly investigate this behavior, we conduct cross-verification experiments using 12 post-trained models (the three smallest models from each of the four families) and compute all verifier metrics for each pair. For intra-family verification, each solver has 2 verifiers from the same family (excluding itself), resulting in $12 \times 2 = 24$ solver–verifier pairs. For cross-family verification, each solver has 9 verifiers from other families, giving $12 \times 9 = 108$ cross-family pairs. For each pair, we plot the verifier metric against the **solver–verifier similarity score**, defined as the average cosine similarity between the two models' solution embeddings across all dataset problems. Solutions are embedded using `sentence-transformers/all-mpnet-base-v2`.

Figure 6 shows that, for both intra-family and cross-family settings, FPR exhibits a clear positive trend: the more similar the solver and verifier are in their solution distributions, the more likely the verifier is to accept the solver's incorrect answers. While intra-family verifier gains are too small to yield a strong correlation, cross-family verifier gains decrease significantly with increasing similarity. This indicates that, when selecting a solver–verifier pair, choosing a verifier whose solution distribution differs from that of the solver leads to more reliable verification.

**Takeaways**

Higher similarity between solver and verifier solution distributions increases the verifier's tendency to accept incorrect solver outputs, reducing verifier gain. Using a verifier with a meaningfully different solution distribution mitigates this bias.

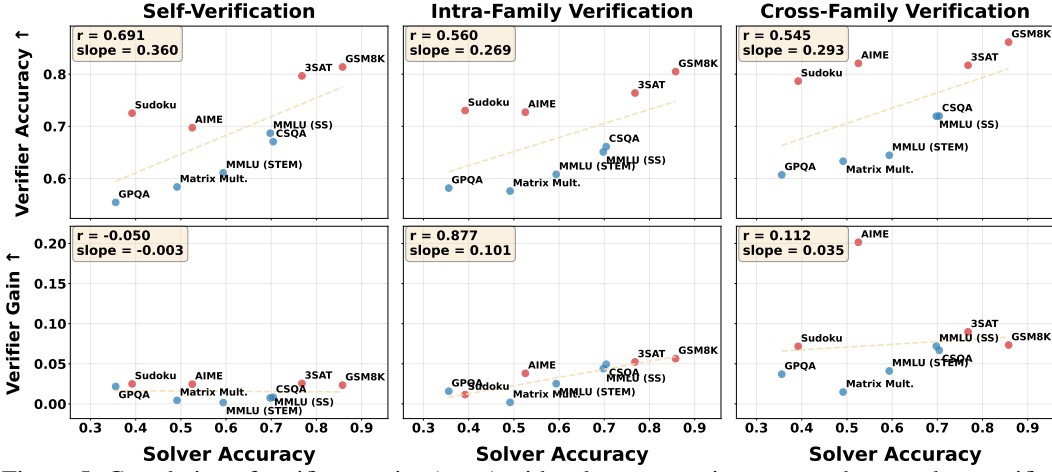

Figure 5: Correlation of verifier metrics (rows) with solver accuracies, averaged over solver-verifier pairs that belong to each verification setting (columns).

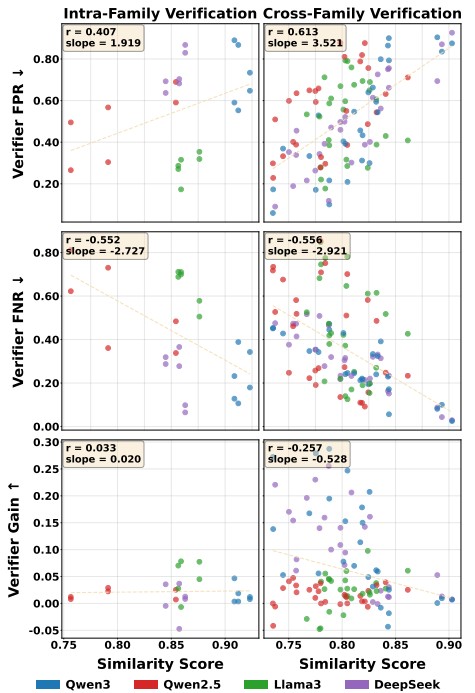

Figure 6: Correlation between verifier metrics with similarity scores between solver-verifier pairs. Each marker is colored based on the verifier model family.

### 5.4 HOW DOES POST-TRAINING AFFECT SOLVER AND VERIFIER PERFORMANCE?

We examine how post-training influences verifier behavior. Our analysis focuses on the `Qwen2.5-Base`/`Qwen2.5` and `Qwen3-Base`/`Qwen3` model pairs. We exclude `Llama3-Base` due to its weak solver and verifier performance and omit `DeepSeek` because matching base models are unavailable. Verification metrics are computed across all 37 base and post-trained models.

**Post-training effect on solver performance.** We begin by evaluating how solver accuracy changes after post-training. For each model pair, we compute solver accuracy and average results across model families and datasets. As expected, post-training yields substantial improvements: `Qwen2.5` solvers improve by an average of 8.2%, while `Qwen3` shows a striking 35.4% gain. Full results are provided in Appendix I.

**Post-training effect on verifier performance.** We next analyze how post-training affects verifier behavior (Figure 7). For each model, we compute verifier metrics against all solvers and datasets, partition results by verification setting, and average within families. For both `Qwen2.5` and `Qwen3`, post-training increases FPR and reduces verifier gain in self-verification, despite improvements in FNR. Although `Qwen3` benefits more than `Qwen2.5` in solver accuracy (Figure 11), its post-trained verifiers show higher FPRs and lower gains in both self- and intra-family verification. In contrast, both families, and especially `Qwen3`, show substantial improvements in cross-family verification.

> **Takeaways**
>
> Post-training significantly enhances a base model's problem-solving ability but can reduce its self- or intra-family improvement potential. In contrast, it boosts models' performance in cross-family verification.

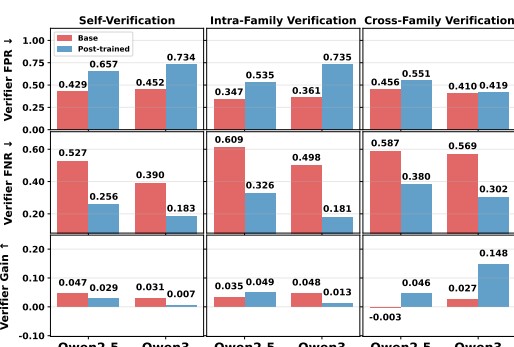

Figure 7: Changes in verifier metrics of the `Qwen2.5-Base` and `Qwen3-Base` models from post-training.

## 5.5 WHICH DATASETS ARE EASY TO VERIFY?

Thus far, we have examined verifier performance and its contribution to solver accuracy through rejection sampling. We now shift to a task-level perspective and ask: *are tasks that are easy to solve also easy to verify?* In Figure 5, we recompute the verifier metrics from Section 5.1, average them over all verifier models, and plot them against solver accuracies.

We find that verification accuracy correlates strongly with solver accuracy. In contrast, verifier gains from self-verification do not correlate with problem difficulty, whereas both intra-family and cross-family verifier gains exhibit clear positive correlations. Notably, AIME appears as an outlier in the final plot, potentially due to models encountering similar problems in post-training.

The best-fit lines for verifier accuracy further reveal two distinct clusters of tasks (colored red and blue), indicating that verification difficulty cannot be explained solely by solver accuracy. This leads to our next question: *are some tasks inherently easier to verify than others?* Relatedly, Song et al. (2025) find that models cannot self-improve on tasks requiring factual recall, but can self-improve on Sudoku with sufficient pretraining scale. We observe a similar separation in Figure 5: AIME, GSM8K, 3SAT, and Sudoku exhibit a higher ratio of verifier accuracy to solver accuracy and deliver higher gains across all verification settings.

To explain this, we notice that among our synthetic datasets, Sudoku and 3SAT are classic examples of problems that require exponential solving time but allow polynomial-time verification. By contrast, there is no clear shortcut for verifying the product of two matrices without effectively recomputing it for Matrix Multiplication. Among the real-world datasets, GSM8K and AIME involve problems solvable with high-school-level mathematics, whereas MMLU (Social Sciences) requires domain-specific knowledge, CommonsenseQA relies on implicit world knowledge, and GPQA and MMLU (STEM) draw on specialized natural science content. For these latter tasks, verifying an answer requires essentially the same knowledge as solving the problem.

> **Takeaways**
>
> Although tasks that are easy to solve are typically easier to verify, some tasks are inherently easier to verify. These include synthetic problems with logical or structured reasoning (e.g., 3SAT, Sudoku) and real-world tasks relying primarily on mathematical reasoning rather than extensive factual recall (e.g., GSM8K, AIME). Such tasks also yield larger gains from test-time rejection sampling with verifiers.

## 6 CONCLUSION

This work presents a comprehensive study of LLM-based verification for problem solving. We show that verification accuracy alone provides an incomplete picture of the expected improvement obtained by using a verifier for test-time rejection sampling, motivating the introduction of *verifier gain*, a more informative measure that captures this expected improvement. Using this metric, our analysis shows that verifier gain is often lower for self-verification and intra-family verification than for cross-family verification, particularly as model size increases or post-training is applied. Further analysis reveals that decreases in verifier gain correlate with greater similarity between the solver's and verifier's solution distributions. Finally, we show that some tasks are inherently easier for LLMs to verify than others: more difficult tasks generally require domain-specific or implicit world knowledge, whereas easier tasks tend to involve logical reasoning, mathematical reasoning, or structured puzzle solving.

**Limitations and future work.** Section 5.5 shows that some tasks are inherently more verifiable than others, motivating future work on developing a predictive model for the verifiability of individual tasks or questions. Section 5.3 shows that LLMs are biased toward accepting incorrect solutions that resemble their own reasoning, indicating that it will be worthwhile to examine the origins of this bias in pre-training and/or post-training.

## 7 REPRODUCIBILITY STATEMENT

We explain our experimental setup in Section 4, provide full lists of models and datasets with their open-source HuggingFace identifiers in Appendix D and Appendix E, respectively. For details on how we compute solver and verifier metrics, we introduce them in Section 3 and provide precise mathematical definitions of them in Appendix B and Appendix C. We submit all inference, evaluation, and synthetic data generation code in the Supplementary Materials with accompanying documentation. All of our experiments are seeded for reproducible results.

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

APPENDIX

## A    ADDITIONAL RELATED WORK

**Scaling test-time compute.** A simple method for scaling test-time compute involves sampling several candidates and selecting one *post hoc*, for example via Best-of-$N$. This can take the form of sample-and-rank approaches (Nichols et al., 2020), majority vote (Wang et al., 2023), model-based aggregation (Chen et al., 2024), or sampling then filtering (Weng et al., 2023). LLMs can also be finetuned to explicitly optimize Best-of-$N$ performance (Chow et al., 2025). Instead of *post hoc* selection, we can guide the model towards good samples via constraints (Roy & Roth, 2015), a scoring function (Yang et al., 2022), or sequential construction (Kang et al., 2024; Khalifa et al., 2023).

## B    ADDITIONAL DETAILS ON VERIFIER METRICS

We show the mathematical definitions of relevant verifier metrics below. For clarity, we include dependencies (e.g., $(S, V; \mathcal{D})$) in the definitions, but sometimes omit them for brevity when the context is clear.

$$\text{VerifierAcc}(S, V; \mathcal{D}) = \mathbb{E}_{(x, \mathcal{Y}_x) \sim \mathcal{D}, \, y \sim S(x)} \big[ \mathbb{1}\{ V(x, y) = c(x, y) \} \big]$$

$$\text{TPR}(S, V; \mathcal{D}) = \mathbb{E}\big[ V(x, y) \mid y \in \mathcal{Y}_x \big]$$

$$\text{FPR}(S, V; \mathcal{D}) = \mathbb{E}\big[ V(x, y) \mid y \notin \mathcal{Y}_x \big]$$

$$\text{FNR}(S, V; \mathcal{D}) = \mathbb{E}\big[ 1 - V(x, y) \mid y \in \mathcal{Y}_x \big] = 1 - \text{TPR}(S, V; \mathcal{D}).$$

$$\text{Precision}(S, V; \mathcal{D}) = \mathbb{E}\big[ c(x, y) \mid V(x, y) = 1 \big] = \frac{\text{SolverAcc} \cdot \text{TPR}}{\text{SolverAcc} \cdot \text{TPR} + (1 - \text{SolverAcc}) \cdot \text{FPR}}$$

$$\text{Recall}(S, V; \mathcal{D}) = \text{TPR}$$

$$\text{F1}(S, V; \mathcal{D}) = \frac{2 \cdot \text{Precision} \cdot \text{Recall}}{\text{Precision} + \text{Recall}}$$

## C    ADDITIONAL DETAILS ON VERIFICATION SETTINGS

We show the mathematical definitions of our three verification settings. Let $\mathcal{M}$ denote the space of models and $\mathcal{F}$ the space of model families. We define a function

$$\text{Family} : \mathcal{M} \to \mathcal{F},$$

that maps each model (e.g., `meta-llama/Meta-Llama-3-70B`) to its corresponding family (e.g., `Llama-3-Base`). Note that $S, V \in \mathcal{M}$. For any verifier metric $M(\cdot, \cdot; \mathcal{D})$ such as Verifier-Acc, TPR, or $G$, we define:

$$\text{Self-Verif}(V; \mathcal{D}, M) = M(V, V; \mathcal{D}),$$

$$\text{Intra-Verif}(V, \mathcal{S}; \mathcal{D}, M) = \frac{\sum\limits_{\substack{S \in \mathcal{S} \\ S \neq V, \, \text{Family}(S) = \text{Family}(V)}} M(S, V; \mathcal{D})}{|\{S \in \mathcal{S} : S \neq V, \, \text{Family}(S) = \text{Family}(V)\}|},$$

$$\text{Cross-Verif}(V, \mathcal{S}; \mathcal{D}, M) = \frac{\sum\limits_{\substack{S \in \mathcal{S} \\ \text{Family}(S) \neq \text{Family}(V)}} M(S, V; \mathcal{D})}{|\{S \in \mathcal{S} : \text{Family}(S) \neq \text{Family}(V)\}|}.$$

## D    ADDITIONAL DETAILS ON MODELS

We show the information for each of our 37 evaluated models in Table 1.

## E    ADDITIONAL DETAILS ON DATASETS

### E.1    REAL-WORLD DATASETS

Note that for MMLU (STEM) and MMLU (Social Sciences), we concatenate questions from all subjects that belong to the STEM and Social Sciences supercategories in Hendrycks et al. (2021), respectively.

## E.2 SYNTHETIC DATASETS

We generate three synthetic datasets, named 3SAT, Matrix Multiplication, and Sudoku, with 1000 samples each. We submit the data generation code in the Supplementary Materials, but briefly explain each synthetic dataset's generation parameters below.

Each 3SAT CNF contains uniformly sampled numbers of variables and clauses from 2 to 8 (inclusive). Each Sudoku puzzle is a 9x9 grid with 12 randomly missing cells. Each Matrix Multiplication problem is about multiplying 2 4x4 integer matrices with values uniformly sampled from $[-5, 5]$. All data are generated in a way that ensures the existence of a valid solution. Note that while Matrix Multiplication has a singular correct answer for each problem, Sudoku and 3SAT are allowed multiple correct answers as long as the solver's answer is correct by their rules.

The generation code files for all synthetic datasets are seeded for reproducibility.

An example of a generated 3SAT problem:

```
## Problem Definition

**SAT (Boolean Satisfiability Problem)** is a fundamental problem
in computer science where we need to determine if there exists an
assignment of Boolean values (True/False) to variables that makes
a given Boolean formula evaluate to True.
**Variables**: In this problem, variables are named as single
letters. Each variable can be assigned either True (T) or
False (F).
**Literals**: A literal is either a variable (like a) or its
negation (like ˜a, meaning "not a"). If a is True, then ˜a is
False, and vice versa.
**Clauses**: A clause is a disjunction (OR operation) of literals.
A clause is satisfied (True) if at least one of its literals is
True. For example, the clause (a or ˜b) is True if either a is
True OR b is False (or both).
**CNF (Conjunctive Normal Form)**: The Boolean formula is given in
CNF, which is a conjunction (AND operation) of multiple clauses.
The entire formula is satisfied only if ALL clauses are satisfied
simultaneously.
**3SAT**: This is a special case of SAT where every clause contains
exactly 3 literals.

## The Problem

Find a satisfying assignment for the following CNF formula:
(˜c or ˜b or d) and (d or ˜b or ˜c) and (d or a or c) and
(˜c or d or a) and (b or ˜a or d) and (c or d or ˜b)

## Instructions

Provide your answer as a list of variable assignments, one per line,
in the format "variable_name T" or "variable_name F". For example:
\boxed{
a T
b F
}
This means a=True, b=False.

Another example answer is
\boxed{
a F
b T
}
```

This means a=False, b=True.

Output and only output the T/F values for the variables that appear in the provided CNF formula.

An example of a generated Sudoku problem:

## Sudoku Problem

**Sudoku** is a logic-based number-placement puzzle. The objective is to fill a 9x9 grid with numbers so that each column, each row, and each of the 3x3 sub-grids contains all of the numbers from 1 to 9.

## The Puzzle

Complete the following 9x9 Sudoku grid (empty cells are marked with '_'):

```
7 4 2 1 _ 5 8 9 6
1 6 9 2 4 8 3 5 7
8 5 3 _ _ 7 2 1 4
2 _ 8 9 7 1 4 6 5
5 7 6 4 8 2 9 3 _
4 9 1 3 _ 6 _ 8 _
3 1 5 8 2 4 6 7 9
6 8 _ 7 1 _ 5 2 3
_ 2 7 5 6 _ 1 4 8
```

## Instructions

Provide your answer as a completed 9x9 grid with all numbers filled in, formatted exactly like the puzzle above but with numbers instead of underscores.

For example, a completed 4x4 grid should look like:
\boxed{
1 2 3 4
3 4 1 2
2 3 4 1
4 1 2 3
}

An example of a generated Matrix Multiplication problem:

## Matrix Multiplication Problem

**Matrix Multiplication** is a fundamental operation in linear algebra where we compute the product of two matrices. For two square matrices A and B of size 4x4, the product C = A x B is computed as:

C[i][j] = Sum(k=0 to 3) A[i][k] x B[k][j]

## The Problem

Compute the product of the following two 4x4 matrices:

**Matrix A:**
```
0 1 1 4
-1 3 4 4
-2 -5 -5 0
```

```
-4 4 5 0

**Matrix B:**
1 2 0 5
1 -2 0 0
3 -1 -3 -3
2 5 -4 2

## Instructions

Provide your answer as the resulting 4x4 matrix C = A x B,
formatted with each row on a separate line and numbers separated
by spaces.

For example, a 2x2 result matrix is formatted like:
\boxed{
1 2
3 4
}
```

## F  ADDITIONAL DETAILS ON EXPERIMENTAL SETUP

We use the following solver prompt for all models:

```
Please reason step by step, and put your final
answer within \boxed{{}}.

{question}
```

We use the following verifier prompt for all models:

```
You are a teacher that is evaluating a student's answer
to a question. Your task is to determine whether the answer
is correct or incorrect.

Question: {question}

Student's Answer: {response}

Please evaluate the student's answer carefully. Consider:
- Is the answer factually accurate?
- Is the reasoning sound and logical?
- Does it fully address the question asked?

After your evaluation, provide your judgment in the
following format:
- If the answer is correct, write: \boxed{{correct}}.
- If the answer is incorrect, write: \boxed{{incorrect}}.

First explain your analysis over the student's answer,
then provide your final judgment in the boxed format.
Make sure the final judgment is either "correct" or
"incorrect" inside the \boxed{{}}. Do not put anything
else in \boxed{{}}. Do not repeat the student's answer
in \boxed{{}}.
```

Figure 8 displays the ratio of filtered solver outputs due to not containing a box for answer extraction, averaged across all datasets.

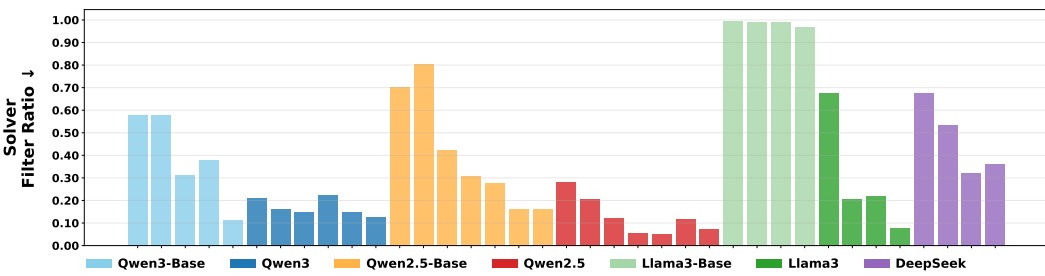

Figure 8: Average ratio of filtered solver outputs for each model over all datasets. Base model families are suffixed by **-Base**. Models within each family are ordered in increasing size.

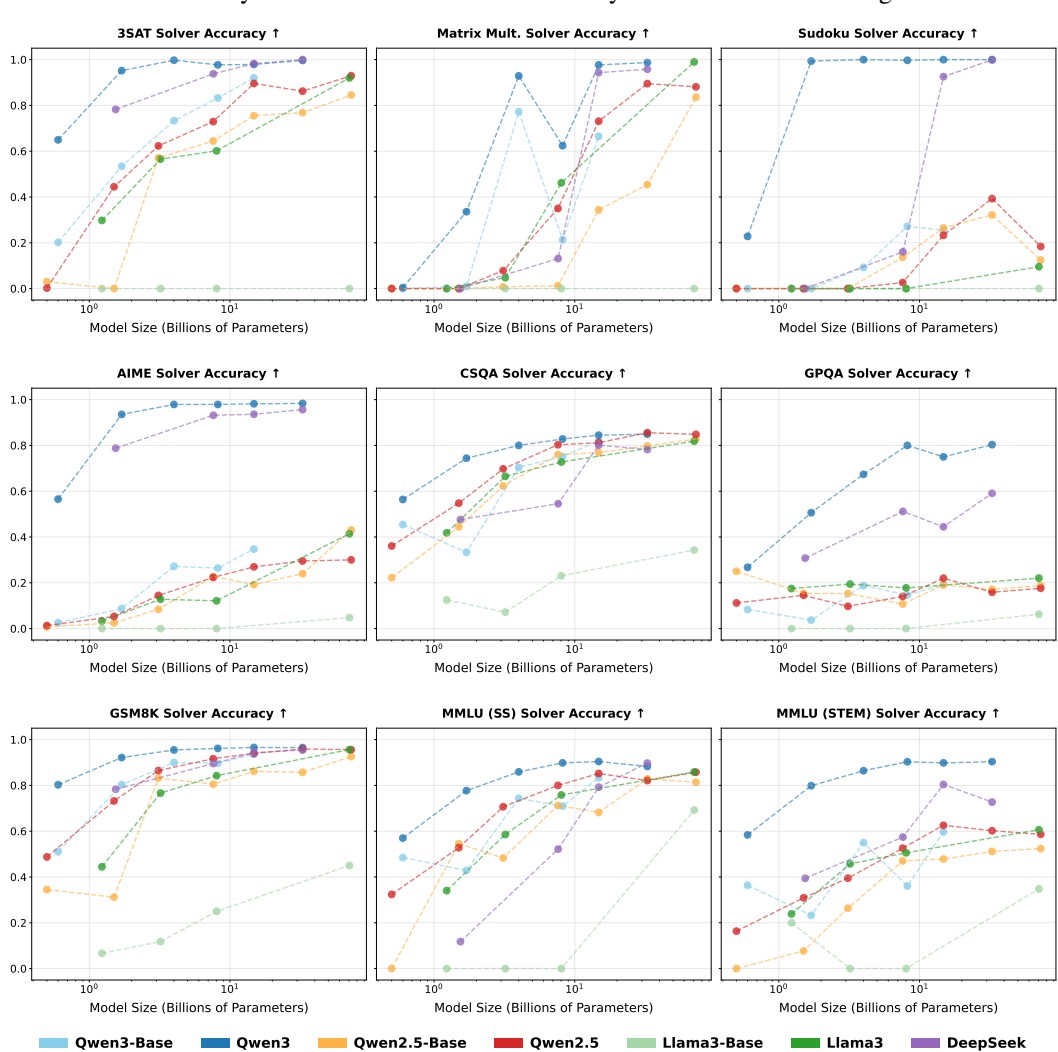

Figure 9: The solver accuracies of 37 models on each dataset.

# G SOLVER ACCURACY BY DATASET

Figure 9 shows the solver accuracies of all 37 models on each of our 9 datasets.

# H F1-SCORE AND PRECISION VISUALIZATION

Figure 2 shows the correlation between each model's verification ability and its own solver accuracy for all 21 post-trained models. We additionally display verifier F1-Score and precision in Figure 10.

In comparison to verifier accuracy, while F1-Score also positively correlates with verifier's own solver accuracy for all verification settings, the slopes decrease from self-verification to intra-family

Table 1: Complete list of each evaluated model's HuggingFace identifier, family, and size.

| HuggingFace Identifier | Family | Size |
|---|---|---|
| Qwen/Qwen3-0.6B-Base | Qwen3-Base | 0.6B |
| Qwen/Qwen3-1.7B-Base | Qwen3-Base | 1.7B |
| Qwen/Qwen3-4B-Base | Qwen3-Base | 4B |
| Qwen/Qwen3-8B-Base | Qwen3-Base | 8B |
| Qwen/Qwen3-14B-Base | Qwen3-Base | 14B |
| Qwen/Qwen3-0.6B | Qwen3 | 0.6B |
| Qwen/Qwen3-1.7B | Qwen3 | 1.7B |
| Qwen/Qwen3-4B | Qwen3 | 4B |
| Qwen/Qwen3-8B | Qwen3 | 8B |
| Qwen/Qwen3-14B | Qwen3 | 14B |
| Qwen/Qwen3-32B | Qwen3 | 32B |
| Qwen/Qwen2.5-0.5B | Qwen2.5-Base | 0.5B |
| Qwen/Qwen2.5-1.5B | Qwen2.5-Base | 1.5B |
| Qwen/Qwen2.5-3B | Qwen2.5-Base | 3B |
| Qwen/Qwen2.5-7B | Qwen2.5-Base | 7B |
| Qwen/Qwen2.5-14B | Qwen2.5-Base | 14B |
| Qwen/Qwen2.5-32B | Qwen2.5-Base | 32B |
| Qwen/Qwen2.5-72B | Qwen2.5-Base | 72B |
| Qwen/Qwen2.5-0.5B-Instruct | Qwen2.5 | 0.5B |
| Qwen/Qwen2.5-1.5B-Instruct | Qwen2.5 | 1.5B |
| Qwen/Qwen2.5-3B-Instruct | Qwen2.5 | 3B |
| Qwen/Qwen2.5-7B-Instruct | Qwen2.5 | 7B |
| Qwen/Qwen2.5-14B-Instruct | Qwen2.5 | 14B |
| Qwen/Qwen2.5-32B-Instruct | Qwen2.5 | 32B |
| Qwen/Qwen2.5-72B-Instruct | Qwen2.5 | 72B |
| meta-llama/Llama-3.2-1B | Llama3-Base | 1B |
| meta-llama/Llama-3.2-3B | Llama3-Base | 3B |
| meta-llama/Llama-3.1-8B | Llama3-Base | 8B |
| meta-llama/Llama-3.1-70B | Llama3-Base | 70B |
| meta-llama/Llama-3.2-1B-Instruct | Llama3 | 1B |
| meta-llama/Llama-3.2-3B-Instruct | Llama3 | 3B |
| meta-llama/Llama-3.1-8B-Instruct | Llama3 | 8B |
| meta-llama/Llama-3.1-70B-Instruct | Llama3 | 70B |
| deepseek-ai/DeepSeek-R1-Distill-Qwen-1.5B | DeepSeek | 1.5B |
| deepseek-ai/DeepSeek-R1-Distill-Qwen-7B | DeepSeek | 7B |
| deepseek-ai/DeepSeek-R1-Distill-Qwen-14B | DeepSeek | 14B |
| deepseek-ai/DeepSeek-R1-Distill-Qwen-32B | DeepSeek | 32B |

verification, and further decrease for cross-family verification, showing that the increase in false positive rate in Figure 2 has a stronger negative impact on lowering F1-score than accuracy.

While Section 5.1 explains the low verifier gains for self- and intra-family verification through close examination of FPR, we additionally plot verifier precision in Figure 10. However, since precision is the expected performance of verifier-based rejection sampling in the limit of infinite sampling and our main metric "verifier gain" is defined in terms of it (Equation 1), precision does not help explain the differences in verifier gains across verification settings itself.

## I  EFFECT OF POST-TRAINING ON SOLVER PERFORMANCE

Figure 11 shows the average improvement in solver accuracies of `Qwen2.5-Base` and `Qwen3-Base` families of models from their respective post-training procedures.

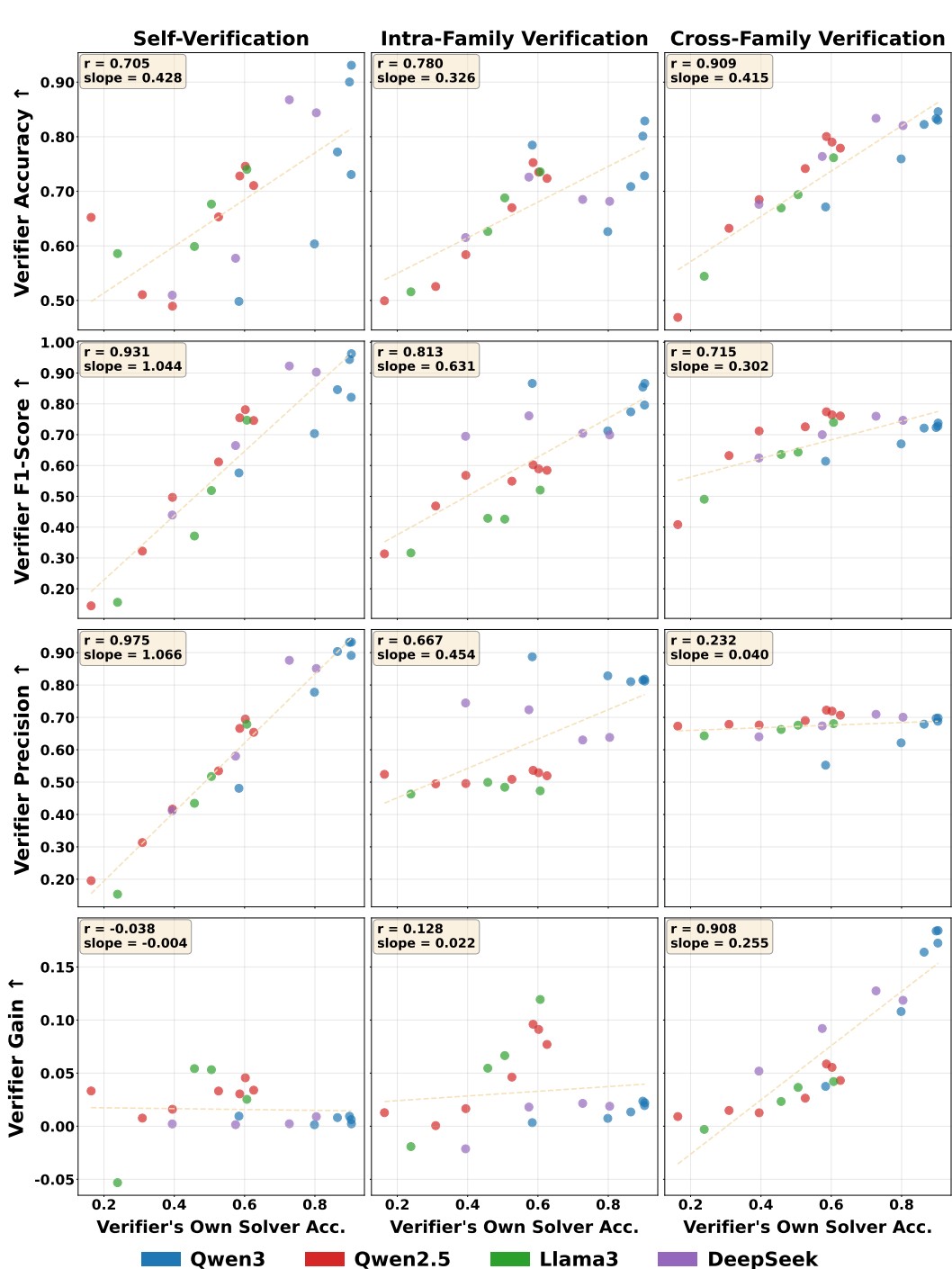

Figure 10: Correlation between each model's verifier metrics (rows) and its own solver accuracy for all 21 post-trained models, averaged over all datasets. Each verifier metric is computed over three settings (columns): self-verification, intra-family verification, and cross-family verification. We use the same set of post-trained models as the set of solver models.

Table 2: HuggingFace information and sizes of real-world datasets.

| Dataset Name | HuggingFace Identifier | HuggingFace Split | Size |
|---|---|---|---|
| GSM8K | openai/gsm8k | test | 1319 |
| AIME | TianHongZXY/aime-1983-2025 | test | 963 |
| MMLU (STEM) | cais/mmlu | test | 316 |
| MMLU (Social Sciences) | cais/mmlu | test | 308 |
| CSQA | tau/commonsense_qa | validation | 2442 |
| GPQA | Idavidrein/gpqa | train | 198 |

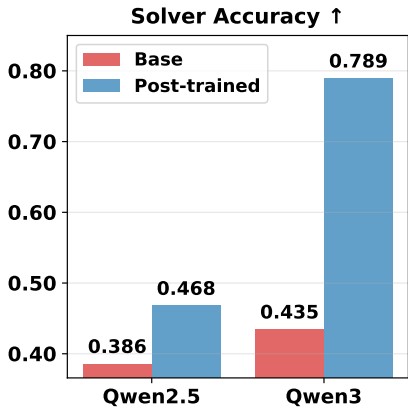

Figure 11: Improvements in solver accuracies of `Qwen2.5-Base` and `Qwen3-Base` models from post-training.

