# OpenReview forum: "Beyond Solving: A Closer Look at LLMs as Solution Verifiers"
_ICLR.cc/2026/Conference — Submitted to ICLR 2026_

### Official Review · Reviewer_Y1Lr · 2025-10-23

**Soundness:** 3
**Presentation:** 3
**Contribution:** 2
**Rating:** 4
**Confidence:** 4

**Summary:**

This paper presents a systematic study on the capability of LLMs to act as solution verifiers. The research investigates solver-verifier interactions across 37 different models—spanning various families, sizes, and post-training methods—on 9 diverse benchmarks, including logic, math, and commonsense reasoning. To measure the impact of verification, the authors introduce "verifier gain," a new metric that predicts the performance improvement achieved by using an LLM to select the best answer from multiple candidates. They found that cross-family verification is often more beneficial than self-verification and intra-family verification. The study analyzes how this verification ability scales with model size and post-training, while also characterizing the inherent "verifiability" of different datasets.

**Strengths:**

This work conducts extensive experiments on solver-verifier interactions and found that verifiers are more helpful when they are from a different family. This is insightful and could benefit future work.
The proposed verifier gain is a good indicator to simulate the gains.

**Weaknesses:**

1. The "verifier gain" metric is proposed but did not explain the reason behind it, which is confusing. In addition, since it is the upper bound, the name may not propriate. For example, if I have a verifier that can achieve 100% precision, the solver-verifier still cannot achieve 100% if the solver is too weak, even with unlimited sampling.
2. The observations are mainly empirical and lacks in-depth analysis. For example, they found that self-verification does not have much gain but did not explain the reason. One possible reason is self-enhancement bias [1]. To explain the reason, analysis of precision (in equation 1) in different conditions should be conducted. Is the precision lower when verifying the model's own responses?
3. Some observations can be anticipated, which may limit the novelty of this work. For example, self-verification is not so effective due to self-enhancement bias.

[1] Jonathon D Brown. Evaluations of self and others: Self-enhancement biases in social judgments. Social cognition, 4(4):353–376, 1986.

**Questions:**

In Figure 2, 5, and 6, what is the meaning of "r = 0.xxx" in the legend?

---

> ### Author Response · Authors · 2025-11-21
>
> We thank the reviewer for their thoughtful and constructive feedback. We appreciate the recognition that our findings are insightful, beneficial for future work, and that verifier gain serves as a useful indicator for verifier performance.
>
> > The "verifier gain" metric is proposed but did not explain the reason behind it, which is confusing. In addition, since it is the upper bound, the name may not propriate. For example, if I have a verifier that can achieve 100% precision, the solver-verifier still cannot achieve 100% if the solver is too weak, even with unlimited sampling.
>
> **Verifier gain and its interpretation as an “upper bound.”** Section 3.2 introduces verifier gain in the context of a rejection-sampling scheme where a solver $S$ repeatedly samples $\hat y\sim S(x)$ until a verifier accepts a solution ($V(x, \hat y)=1$) or we hit a budget. Assuming that the solver assigns non-zero probability to at least one correct solution per problem (we revised Section 3.2 to make this assumption explicit), the correctness of the accepted solution clearly converges to the verifier’s precision in the limit of infinite resampling. Equation 1 then defines verifier gain as precision minus solver accuracy, which is exactly the “upper bound” improvement that a verifier can theoretically enable under such rejection sampling, relative to using the solver alone.
>
> **Verifier gain’s motivation and usage.** Our goal is to compare verifiers relative to one another for a given solver. Throughout Sections 5.1-5.2, we use verifier gain in this comparative sense, rather than as a realized improvement. Section 5.2 further shows that verifier gain correlates strongly with empirical improvements under finite (e.g., 9) attempts, validating its usefulness as a predictive metric even though realized gains are naturally smaller under realistic budgets.
>
> **The name “verifier gain.”** The reviewer correctly points out that verifier gain is an “upper bound”, and we have revised Section 3.2 to clarify that verifier gain is an **asymptotic and comparative metric**. As the reviewer notes, highly biased or weak solvers may not reach this bound in practice.
>
> > The observations are mainly empirical and lacks in-depth analysis. For example, they found that self-verification does not have much gain but did not explain the reason. One possible reason is self-enhancement bias [1]. To explain the reason, analysis of precision (in equation 1) in different conditions should be conducted. Is the precision lower when verifying the model's own responses?
>
> **Incorporating self-enhancement bias.** We thank the reviewer for pointing us to self-enhancement bias. We agree that human psychology can motivate our intuitions surrounding LLM behavior, and Section 5.3 actually mentions a similar intuition: “When a human is asked to cross-check a solution that resembles their own reasoning, they may think it’s more likely to be true.” We revised Section 5.3 to explicitly reference self-enhancement bias. However, while such human analogies can offer helpful intuition, they are not sufficient to conclusively explain LLM behavior without direct empirical evidence.
>
> **Explaining limited self-verification verifier gain.** The investigation of the reasons behind self-verification’s verifier gain is the main goal of Sections 5.1 and 5.3.
> 1. Section 5.1 / Figure 2: FPR increases with a verifier’s own problem-solving strength during self- and intra-family verification where gains are weak, but not during cross-family verification where gains are strong.
> 2. Section 5.3 / Figure 6: FPR grows with solver-verifier similarity for both intra-family and cross-family verification, and verifier gain decreases as this similarity increases.
>
> Putting them together, these results show that the more similar solver and verifier solution distributions are (exactly the same distribution in the case of self-verification), the higher the verifier’s FPR, leading to lower verifier gain. This can be viewed as an “LLM analogue” of the self-enhancement bias in humans.
>
> **Why precision is not the right diagnostic for explaining verifier gain.** We agree that precision is an important metric and, in fact, we state in Section 3.2 that precision is the expected performance of verifier-based rejection sampling in the limit of infinite sampling, and define our main metric “verifier gain” in terms of it:
> $$
> \text{Gain}(S,V;\mathcal{D}) = \text{Precision}(S,V;\mathcal{D}) - \text{SolverAcc}(S;\mathcal{D}).
> $$
> However, precision is already by definition the baseline solver accuracy plus verifier gain. When verifier gain is low, precision is close to the baseline solver accuracy, so simply plotting precision does not explain why verifier gain itself is low. We have revised Appendix H to include precision plots alongside verifier gains, showing that precision is not consistently lower for self-verification and that it does not help in understanding the reason behind self-verification’s low gains.

---

> > ### Author Response · Authors · 2025-11-21
> >
> > > Some observations can be anticipated, which may limit the novelty of this work. For example, self-verification is not so effective due to self-enhancement bias.
> >
> > While some of our conclusions are consistent with expectations one might have for LLMs based on human cognition (self-enhancement bias), our goal is not just to restate them, but to turn them into quantitative, sometimes counterintuitive guidance for choosing verifiers in practice.
> >
> > To highlight where our findings go beyond expectations, here are several phenomena that were **not obvious** to us prior to running the experiments and, to our knowledge, have not been documented in prior work:
> > 1. Modern post-trained models exhibit almost no self-verification gain, even though their solver accuracy is high. In contrast, their cross-family verification ability continues to scale (Section 5.1).
> > 2. We show that raw verification accuracy can be misleading for assessing whether a verifier can improve a solver’s accuracy from rejection sampling. We introduce verifier gain, a robust empirical predictor of the improvement from rejection sampling, which lets practitioners compare verifiers without running full rejection-sampling experiments (Sections 5.1–5.2).
> > 3. We show that low verifier gains depend strongly on solver-verifier similarity, revealing that verifiers whose distributions resemble the solver’s can significantly harm verifier gain. This leads to concrete, nontrivial guidance favoring distributionally different verifiers rather than the strongest (in accuracy) or most similar ones (Section 5.3).
> > 4. We show that post-training decreases self- and intra-family verifier gains even as the verifier’s own solving accuracy improves, yet greatly enhances cross-family verification (Section 5.4).
> >
> > These findings offer specific rules of thumb for how to choose verifiers in practice, and expose interaction effects (e.g., solver-verifier similarity) that are difficult to anticipate without a large-scale empirical analysis.
> >
> > > Figure 2, 5, and 6, what is the meaning of "r = 0.xxx" in the legend?
> >
> > “r = 0.xxx” denotes the Pearson correlation coefficient between the plotted x- and y-axes, summarizing the strength of their linear relationship. We include it to help readers quickly assess the robustness of trends.
> >
> > ### **Concluding Note**
> >
> > We thank you again for the thoughtful feedback. We have addressed your concerns about the verifier gain metric, the depth of our analysis for verifier bias, and the novelty of our contributions. Given the extensive scope of our study and the strengthened explanations provided here, we hope you will consider raising your score to reflect the practical contributions and insights offered by this work.

---

> > > ### Comment · Reviewer_Y1Lr · 2025-11-27
> > >
> > > Thanks for the clarification. Even though verification may help improve the accuracy, it may not be practically useful, as it may incur addition cost. If you have access to a strong verifier, why not use it as a solver directly? Or ensemble the responses of solver and verifier?

---

> > > > ### Author Response · Authors · 2025-11-29
> > > >
> > > > We appreciate your engagement and the thoughtful questions about verification.
> > > >
> > > > **Verification is an established and effective way to improve solution quality.** We agree that verification incurs additional costs, so the key question becomes: *How should this additional compute be used most effectively?* A large body of prior work already chooses to incur this cost: [1] shows that verifier-based filtering can outperform large increases in solver model size, [2] performs iterative self-improvement using verifiers, and [3] demonstrates remarkable scalability as compute is increased for both solvers and verifiers. Therefore, verification has already become a widely used technique for improving solution quality, much like self-consistency or ensembling. Our paper directly targets this established setting by asking: *Given that practitioners already spend additional compute on verification, how can they obtain the maximum benefit from it?* We provide an extensive study of self-, intra-family, and cross-family verification and conclude with actionable takeaways on how to use verifiers most effectively.
> > > >
> > > > **Verification offers distinct advantages over solving and ensembling.** Regarding the question “*why not simply use the strong verifier as a solver (or ensemble its solutions with the solver)*”: the solver is fixed in many realistic scenarios (e.g., a production assistant) and the verifier is a separate component selected or trained specifically for judgment, not for solving. In addition, our results show when and why this separation is beneficial: in Section 5.5, we show that for logical reasoning, puzzle solving, and mathematical tasks (e.g., AIME, 3SAT), verification is substantially easier than solving. In contrast, tasks relying on broad factual or domain knowledge (e.g., CSQA, GPQA) do not exhibit this pattern. Thus, in many practical settings, a verifier can reliably recognize correct solutions it could not consistently generate itself, so using it as a solver or ensembling its generated outputs would underutilize the distinct advantage that verification provides.
> > > >
> > > > We hope this response properly clarifies how verification is an effective and well-established use for improving solution quality, as well as the distinct benefits verification can provide. Thank you again for the thoughtful questions.
> > > >
> > > > [1] Cobbe et al., “Training Verifiers to Solve Math Word Problems”, 2021
> > > >
> > > > [2] Song et al., “Mind the Gap: Examining the Self-Improvement Capabilities of Large Language Models”, 2024
> > > >
> > > > [3] Zhao et al., “Sample, Scrutinize and Scale: Effective Inference-Time Search by Scaling Verification”, 2025

---

### Official Review · Reviewer_VDYQ · 2025-10-26

**Soundness:** 3
**Presentation:** 2
**Contribution:** 3
**Rating:** 6
**Confidence:** 5

**Summary:**

This paper studies the relationship between solver-verifier interactions across 37 models and 9 benchmarks. More specifically, the authors study a verifier's ability to correctly judge a solver's outputs, comparing self-verification with verficiation of same-family and corss-family models.

In terms of methodology, the authors propose the verifier gain metric: Prec(SVD) - SolverAcc(SD). Moreover, they group models according to their size and pre-trained/post-trained. Then each solver-verifier pair can be categorized into 1) self-verification, 2) intra-family verification, and 3) cross-family verification. 21 post trained and 16 pre-trained models from 0.5-72B are compared on 9 datasets.

The first resuls show that solver accuracy improves with model capacity, especially post-trained models, and similarly, verifier accuracy improves with verifier model's own solver accuracy. While those are unsurprising, the authors identify less gain during self-verification and models that are more accurate solvers are not better at self-improvement, with the largest gap among cross-family verification. I think this was expected but it is good to have an experimental result to confirm it. Additionnally, the authors show that using the verifier gain is a good predictor for the performance improvement when using a verifier for rejection sampling.

The second set of results show that it is worth it to use a verifier with different solutio ndistributions than the solver which is a good finding. Moreover, and unsurprisingly, post-trained models improve solver accuracies. Interestingly, the same is not true for verifiers: a post-trained model can decrease its ability to improve itself or models from its own family, but it improves its capability as a cross-family verifier. Finally, the authors show that some tasks are inhenretly easier to verify than others.

Overall, this paper is motivated and confirm common (and potentially new for some readers) hypotheses that researchers had regarding solvers/verifiers. I appreciate that the authors answers those doing detailed experiments. I think one difficulty of this paper is the presentation: many plots, analysis, and text. I would encourage the authors to maybe aggregate all their results in one table or charts to simplify the take-away messages (although I acknolwedge the presence of take-away sections but I would go even one step further).

**Strengths:**

- Sound experimental design to confirm hypotheses about solvers and verifiers.
- Many models and lot of intra/cross-family experiments.

**Weaknesses:**

- On one hand, it is good to confirm experimentally intuitions that researchers may have already had. On the other one, it may sound rather incremental.
- Presentation could be improved.

**Questions:**

What would be your general guidelines when starting a new project involving solvers and verifiers?

---

> ### Author Response · Authors · 2025-11-21
>
> We thank the reviewer for their thoughtful feedback, particularly for the comprehensive summary. We appreciate the recognition of our sound experimental design and the scale of this project. As an academic lab with limited compute, we invested substantial effort to evaluate 37x37 solver-verifier pairings across 9 datasets, but we believe this scale is essential for obtaining robust insights that can benefit both researchers and practitioners. We thoroughly address the concerns below.
>
> > On one hand, it is good to confirm experimentally intuitions that researchers may have already had. On the other one, it may sound rather incremental.
>
> While some high-level intuitions may feel intuitive to experienced researchers, our goal is not just to restate those intuitions, but to turn them into quantitative, sometimes counterintuitive guidance for how to actually choose verifiers in practice.
>
> To highlight where our findings go beyond existing assumptions, here are several phenomena that were **not obvious** to us prior to running the experiments and, to our knowledge, have not been documented in prior work:
> 1. Modern post-trained models exhibit almost no self-verification gain, even though their solver accuracy is high. In contrast, their cross-family verification ability continues to scale (Section 5.1).
> 2. We introduce verifier gain, and show that, unlike raw verification accuracy, it is a robust empirical predictor of improvements from rejection sampling, enabling practitioners to compare verifiers without running full rejection sampling experiments (Section 5.1, 5.2).
> 3. We show that low verifier gains depend strongly on solver-verifier similarity, revealing that verifiers whose distributions resemble the solver’s can significantly harm verifier gain. This leads to concrete, nontrivial guidance favoring distributionally different verifiers rather than the strongest (in accuracy) or most similar ones (Section 5.3).
> 4. We show that post-training decreases self- and intra-family verifier gains even as a verifier’s own solving accuracy improves, yet greatly enhances cross-family verification (Section 5.4).
>
> These findings offer specific rules of thumb for how to choose verifiers in practice, and expose interaction effects (e.g., solver-verifier similarity) that are difficult to anticipate without a large-scale empirical analysis.
>
> > I think one difficulty of this paper is the presentation: many plots, analysis, and text. I would encourage the authors to maybe aggregate all their results in one table or charts…
>
> > Presentation could be improved.
>
> We appreciate the reviewer’s feedback and believe clarity is especially important given the breadth of our experiments.
>
> Due to the scale of our study (37x37 solvers/verifiers x 9 datasets), aggregating all our results into one plot became either visually overwhelming or required averaging away the distinctions that ultimately produce our key insights (e.g., loss of the clear separation between self-, intra-, and cross-family trends). To preserve interpretability, we instead structured Section 5 into modular subsections, each with plots for specific purposes and explicit takeaways. This allows readers to connect a finding directly to the figure that demonstrates it, without needing to navigate a dense composite plot.
>
> We also considered adding an additional global-summary table, but due to page constraints, this would have required removing either:
> 1. the modular takeaway statements in Section 5 (hurting readability and traceability), or
> 2. crucial explanatory text and figure captions (hurting reproducibility and clarity).
>
> Given these tradeoffs, we chose the current structure to maximize clarity under the page limit. However, we are certainly interested in incorporating additional presentation suggestions if the reviewer has any.

---

> > ### Author Response · Authors · 2025-11-21
> >
> > > What would be your general guidelines when starting a new project involving solvers and verifiers?
> >
> > Our results suggest the following practical checklist:
> > 1. **Check whether the task is easier to verify than to solve.** As shown in Section 5.5, tasks involving logical or mathematical reasoning often yield higher verifier gains, whereas knowledge-recall tasks (e.g., factual QA, domain-specific problems) may offer little benefit from verification relative to simply using the solver.
> > 2. **Use verifier gain, not accuracy, to evaluate a solver-verifier pair.** Section 5.1 shows that verification accuracy can be misleading for assessing whether a verifier can improve a solver’s accuracy from rejection sampling, while Section 5.2 demonstrates that verifier gain reliably predicts the actual boost from rejection sampling.
> > 3. **Prefer verifiers that “think differently” from the solver.** Section 5.3 shows that solution-distribution similarity increases false positives and reduces gains. Practitioners should therefore prefer verifiers from different model families or training distributions than the solver.
> > 4. **Avoid using strong reasoning models as their own verifiers.** Section 5.1 demonstrates that state-of-the-art models such as Qwen-3 and DeepSeek achieve minimal self-verification gain, despite being strong solvers.
> >
> > We hope this high-level guidance is useful both conceptually and practically.
> >
> > ### **Concluding Note**
> >
> > We thank you again for your thoughtful feedback. We have addressed the concerns about our contribution, explained the strengths of the current presentation, and provided practical guidelines for solver-verifier projects. Given the extensive scope of our study, the improved clarity of our experiments, and the strengthened explanations provided here, we hope you will consider raising your score to reflect the practical contributions and insights offered by this work.

---

### Official Review · Reviewer_hm9P · 2025-10-31

**Soundness:** 2
**Presentation:** 3
**Contribution:** 2
**Rating:** 4
**Confidence:** 3

**Summary:**

This paper explores the effectiveness of using large language models (LLMs) for verification in problem solving. The study examines the interactions between solvers and verifiers across 37 models belonging to intra and cross family, covering various benchmarks such as logical reasoning, puzzles, symbolic computation, mathematical problem solving, commonsense reasoning, and domain knowledge. The authors of the paper found that verification accuracy alone does not provide a complete picture of the expected improvement from rejection sampling using a verifier. They propose a new metric called verifier gain, which better characterizes the improvement. They found that gains are often lower for self-verification and intra-family verification than for cross-verification, especially as model size increases or post-training is applied. The decrease in verifier gain is correlated with an increase in the similarity between the solution distributions of the solver and verifier. The authors also found that some tasks are inherently easier to verify with an LLM than others, with easier tasks involving logical reasoning, mathematical reasoning, or structured puzzle solving, while more difficult tasks require domain-specific knowledge or implicit knowledge about the world.

**Strengths:**

•	This paper explores the interaction between solver and verifiers across 37 models belonging to intra and cross model family which is a significant effort.

•	The paper proposes a new metric called verifier gain to study a verifier’s ability to correctly judge a solver’s outputs.

**Weaknesses:**

- While the paper provides a comprehensive study of LLM-based verification for problem solving, overall novelty is somewhat limited.

**Questions:**

- Line number 287-289 says “As we move to intra-family verification we begin to see more gains, but cross-family verification shows the greatest potential for improvement from a verifier.”  How this gain is related with a) the data used for post-training of the specific cross family model and b) individual evaluation data ?
- The clarity of Figure 6 is somewhat lacking. In particular, I am curious about the rationale behind the number of points (different colored dots) displayed for Cross-Family Verification. Could you provide an explanation for this and specify the total number of points involved?
- How about comparison with baseline solver-verifier methods? Can you please elaborate on this?

---

> ### Author Response · Authors · 2025-11-21
>
> We thank the reviewer for their thoughtful and constructive feedback. We appreciate the recognition of the substantial experimental effort and the usefulness of the verifier gain metric. We thoroughly address the concerns below and revise the PDF accordingly.
>
> > While the paper provides a comprehensive study of LLM-based verification for problem solving, overall novelty is somewhat limited.
>
> We have a number of points of novelty. Our work provides the first systematic, controlled study of solver-verifier interactions across self-, intra-family, and cross-family regimes; introduces and validates verifier gain as a more faithful predictor of verifier-based improvements; analyzes scaling trends with respect to model size and post-training; diagnoses failure modes through a solver-verifier similarity metric; and characterizes dataset-level verifiability patterns. To our knowledge, none of these dimensions has been systematically examined before.
>
> Importantly, our analyses yield actionable guidance on how to choose verifiers for deployed LLM systems:
> 1. Verifier gain is a more reliable metric than verifier accuracy, and cross-family verifiers generally outperform self- and intra-family ones (Sections 5.1-5.2).
> 2. LLM verifiers are more likely to accept incorrect solutions when the solver produces outputs that resemble the verifier’s own solution distribution, which is the main reason behind the limited verifier gain in self- and intra-family verification (Section 5.3).
> 3. Post-training improves cross-family verification but weakens self-verification verifier gains (Section 5.4).
> 4. Logical reasoning, structured puzzle solving, and mathematical reasoning problems benefit more from verifier-based rejection sampling (Section 5.5).
>
> > Line number 287-289 says “As we move to intra-family verification we begin to see more gains, but cross-family verification shows the greatest potential for improvement from a verifier.” How this gain is related with a) the data used for post-training of the specific cross family model and b) individual evaluation data ?
>
> Verifier gain is defined strictly as a property of a solver-verifier-dataset triple (Equation 1). The statement in Lines 287–289 summarizes an empirical trend from Figure 2, where we evaluate 21 post-trained models as solvers or verifiers over 9 datasets ($21 × 21 × 9$ runs).
>
> **(a) Relation to post-training data.** While we are not entirely sure what you mean by asking about the relationship to “the data used for post-training”, we attempt to provide a satisfactory answer: we do not have access to the proprietary post-training data or reward models of Qwen, DeepSeek, or Llama, so we avoid making claims about whether particular models have been exposed to our evaluation problems (or similar problems). Instead, we mitigate these concerns by:
> 1. **Using a very broad evaluation suite and many models.** Our study spans 37 models across major model families and 9 datasets. Because verifier metrics aggregate over large numbers of solver-verifier-dataset combinations, any incidental exposure to a particular dataset during post-training can only have a limited influence (e.g., Figure 2). We believe that a study at this scale is essential for drawing stable, model- and dataset-agnostic conclusions about verifier behavior.
> 2. **Including synthetically generated datasets.** Our programmatically generated datasets (e.g., 3-SAT) are unlikely to appear exactly in models’ post-training data. In Figure 5, we plot verifier metrics for each dataset separately and confirm observations like “cross-verification shows the greatest verifier gain” across these datasets.
>
> **(b) Relation to individual evaluation data.** Although Figure 2 reports metrics averaged across datasets, we perform dataset-level analysis in Section 5.5. We show that while the magnitude of verifier gains varies substantially across tasks, cross-family verification remains the most effective across all datasets.
>
> If we misunderstood anything about your question, please let us know, and we will clarify our response.

---

> > ### Author Response · Authors · 2025-11-21
> >
> > > The clarity of Figure 6 is somewhat lacking. In particular, I am curious about the rationale behind the number of points (different colored dots) displayed for Cross-Family Verification. Could you provide an explanation for this and specify the total number of points involved?
> >
> > We thank the reviewer for the clarifying question. Figure 6 is computed using 12 post-trained models (3 from each of the 4 families). Each dot corresponds to a distinct solver-verifier pair, averaged across 9 datasets and colored by the verifier family. For **intra-family verification**, each of the 12 solvers has 2 corresponding verifiers in its family that are not itself. For **cross-family verification**, each of the 12 solvers has 9 corresponding verifiers from a different family. Therefore, we plot $12 \times 2=24$ dots for intra-family plots and $12 \times 9=108$ dots for cross-family plots. We revised Section 5.3 to include this clarification.
> >
> > > How about comparison with baseline solver-verifier methods? Can you please elaborate on this?
> >
> > We hope to clarify that our goal is not to produce a new solver-verifier method, but rather to study solver-verifier interactions and answer the practical question: given a solver, which verifier yields the greatest test-time improvement?
> >
> > To ensure our analysis reflects the strongest available setup, we adopt the standard CoT-based verification scheme, in which the verifier sees both the solver’s reasoning and final answer and performs its own CoT before outputting a judgment. This approach follows prior work like Mind the Gap [1], which compares CoT verification (ours) against multiple-choice verification, and finds CoT substantially more performant and robust. By using the strongest available and widely adopted verification method, our experiments isolate the effects of solver-verifier interaction, rather than artifacts of weaker verification schemes.
> >
> > ### **Concluding Note**
> >
> > We thank you again for your constructive feedback. We have addressed the novelty concern, clarified how verifier gain relates to post-training data and individual tasks, improved our explanation of Figure 6, and clarified our methodological choices. Given the extensive scope of our study, the improved clarity of our experiments, and the strengthened explanations provided here, we hope you will consider raising your score to reflect the practical contributions and insights offered by this work.
> >
> > [1] Song et al., “Mind the Gap: Examining the Self-Improvement Capabilities of Large Language Models”, 2025

---

> > > ### Comment · Reviewer_hm9P · 2025-11-24
> > > **Response to Authors Comments**
> > >
> > > Thanks for taking time to respond. I have gone through your comments and things are clear now. I feel the current rating is apt for this revised manuscript.

---

### Official Review · Reviewer_ibxX · 2025-11-01

**Soundness:** 3
**Presentation:** 3
**Contribution:** 2
**Rating:** 2
**Confidence:** 3

**Summary:**

The paper presents a comprehensive study o LLM-based verification for problem solving. The paper finds that verification accuracy paints an in complete picture of the expected improvement from rejection sampling from a solver using a verifier, and derive and validate a measure called verifier gain to better characterize that improvement. The paper presents extensive experiments on LLM validation.

**Strengths:**

This paper conducts extensive experiments on using LLMs as verifiers, and from an experimental perspective, the evaluation is quite thorough.
The proposed concept of verifier gain is well-motivated and reasonably supported by the experimental results.

**Weaknesses:**

First, I do not find the motivation of this paper convincing, like the statement:
>Yet, verification with LLMs remains underexplored. Prior work has mainly examined scenarios where solver and verifier are the same model or from the same family; far less is known about how verifiers behave when judging outputs from other families.

However, many models, even from quite early on, have extensively used LLM-based verification during the post-training stage. For example, models such as Kimi K2, Qwen-3, and Minimax-M2 all employ rubric-based approaches using LLMs for verification, which also include the so-called cross-family validation.


Second, a large portion of the experiments in this paper are conducted on verifiable tasks such as GSM8K, AIME, and GPQA. I do not believe these experiments have much practical value, as I fail to see any real benefit of using verifiers in production settings for tasks that are already verifiable.

Moreover, the majority of the experiments are carried out on open-source models. Why not test on more capable closed-source models such as GPT-5, Gemini, or Claude?

Finally, the contribution of the paper is somehow trivial. Many experiments remain at surface-level observations. For instance, the authors claim that “verifier models are biased toward accepting incorrect solutions during self-verification or intra-family verification”? What is the underlying reason? And how should experiments be designed to validate this claim?

**Questions:**

See Above

---

> ### Author Response · Authors · 2025-11-21
>
> We thank the reviewer for their thoughtful and constructive feedback. We are encouraged that the reviewer finds our evaluation of LLMs as verifiers thorough and the concept of verifier gain well-motivated. We thoroughly address the concerns below.
>
> > First, I do not find the motivation of this paper convincing… However, many models, even from quite early on, have extensively used LLM-based verification during the post-training stage. For example, models such as Kimi K2, Qwen-3, and Minimax-M2 all employ rubric-based approaches using LLMs for verification, which also include the so-called cross-family validation.
>
> While modern LLMs incorporate LLM-based judging during post-training (we note “verifiers can be used as reward models **during reinforcement learning**” in our Introduction), our work studies a different and complementary question: how do different verifiers behave **during inference time**, and which verifier should one choose to improve solver accuracy? This distinction leads to various key differences:
>
> 1. **Post-training verification ≠ inference-time verification.** Models such as Kimi K2, Qwen-3, and Minimax-M2 employ LLM judges during RL to shape model behavior, but this setting differs fundamentally from inference-time solver-verifier interactions, where the ground truth is unknown. Our focus is on test-time performance improvement through verifier-based rejection sampling, requiring metrics such as verifier gain (Section 3.2) and analyses of FPR/FNR (Section 5).
> 2. **Existing post-training pipelines do not study different verifier behaviors.** For instance, the Qwen-3 report specifies the use of Qwen2.5-72B-Instruct as a reward model, but neither ablates different verifier choices nor measures verifier quality. In contrast, a major question our paper asks is which LLM-based verifier one should choose. To answer this, we study the behaviors of 37 verifiers across 9 tasks and analyze their behavior with respect to solver/verifier family, size, and training stage.
> 3. **No prior work performs controlled experiments across our verification settings.** To our knowledge, no prior work studies the differences in verifier behavior between self-, intra-family, and cross-family verification or the effect of post-training on verification behavior. In contrast, our controlled study evaluated every solver-verifier pair under identical sampling parameters and datasets, enabling us to isolate failure modes such as false-positive bias (Section 5.1) and distribution similarity (Section 5.3).
>
> > Second, a large portion of the experiments in this paper are conducted on verifiable tasks such as GSM8K, AIME, and GPQA. I do not believe these experiments have much practical value, as I fail to see any real benefit of using verifiers in production settings for tasks that are already verifiable.
>
> We appreciate the concern and highlight that our setting differs from post-training verification. At training time, LLM-based verification of tasks like AIME is indeed redundant because labels exist. Our work instead focuses on **scaling test-time compute** with verifier-based rejection sampling. In production settings, users frequently pose new reasoning questions, but the system cannot rely on access to ground-truth answers. A verifier that accurately judges solver outputs enables rejection sampling to boost correctness.
>
> Moreover, verifiable tasks enable evaluation with precise, objectively correct labels. Non-verifiable tasks suffer from subjective or noisy labels that are often produced by human evaluation. While we believe that non-verifiable tasks represent an important research domain, they are unsuitable for our controlled comparative study of verifiers.

---

> > ### Author Response · Authors · 2025-11-21
> >
> > > Moreover, the majority of the experiments are carried out on open-source models. Why not test on more capable closed-source models such as GPT-5, Gemini, or Claude?
> >
> > We appreciate the suggestion. Several practical and scientific considerations guided our decision to focus on open-source models:
> > 1. **Reproducibility.** Closed-source model APIs evolve silently and may not maintain stable behavior, making replication impossible. By focusing on open-source models, we ensure that every number in our paper can be replicated reliably. This project is compute-intensive for an academic lab, so we will open-source all code and raw data for the community. Moreover, closed-source licenses may also prohibit the redistribution of model outputs.
> > 2. **Access to reasoning traces.** Our evaluation assumes the verifier has access to the solver’s intermediate reasoning steps, which reflects practical deployment settings where step-level reasoning improves verification performance. Closed-source models typically do not expose reasoning traces, making them incompatible with our setting.
> > 3. **Need for access to model sizes and training stages.** A major contribution of our paper is analyzing how verification performance differs between model sizes and training stages. Closed-source models often do not release model sizes or base variants, making such analyses infeasible.
> >
> > > Finally, the contribution of the paper is somehow trivial. Many experiments remain at surface-level observations. For instance, the authors claim that “verifier models are biased toward accepting incorrect solutions during self-verification or intra-family verification”? What is the underlying reason? And how should experiments be designed to validate this claim?
> >
> > Our work provides the first systematic, controlled study of solver-verifier interactions across self-, intra-family, and cross-family regimes; introduces and validates verifier gain as a more faithful predictor of verifier-based improvements; analyzes scaling trends with respect to model size and post-training; diagnoses failure modes through a solver-verifier similarity metric; and characterizes dataset-level verifiability patterns. To our knowledge, none of these dimensions has been systematically examined before.
> >
> > Importantly, our analyses yield actionable guidance on how to choose verifiers for deployed LLM systems:
> > 1. Verifier gain is a more reliable metric than verifier accuracy, and cross-family verifiers generally outperform self- and intra-family ones (Sections 5.1-5.2).
> > 2. LLM verifiers are more likely to accept incorrect solutions when the solver produces outputs that resemble the verifier’s own solution distribution, which is the main reason behind the limited verifier gain in self- and intra-family verification (Section 5.3).
> > 3. Post-training improves cross-family verification but weakens self-verification verifier gains (Section 5.4).
> > 4. Logical reasoning, structured puzzle solving, and mathematical reasoning problems benefit more from verifier-based rejection sampling (Section 5.5).
> >
> > > the authors claim that “verifier models are biased toward accepting incorrect solutions during self-verification or intra-family verification”? What is the underlying reason? And how should experiments be designed to validate this claim?
> >
> > In Section 5.3, our explanation for this bias begins with the intuition that humans tend to trust reasoning resembling their own. We hypothesize that verifiers are more likely to accept incorrect solutions whose reasoning resembles their own distribution. We validate this in Figure 6: we compute solver-verifier similarity by embedding both models’ generated solutions, averaging the cosine similarity, and then examining how this score relates to FPR, FNR, and verifier gain. The trends are separated into intra-family and cross-family verification, showing that higher similarity predicts higher false-positive rates and weaker gains in both settings, providing a direct quantitative explanation of the observed bias.
> >
> > ### **Concluding Note**
> >
> > We appreciate your engagement and hope you will consider raising your score to reflect the rigor, scope, and practical contributions of this work. We have addressed all concerns regarding our motivation, our distinction from using verifiers for post-training, our model and task choices, and the underlying causes of verifier bias. Our paper offers meaningful, nontrivial insights into how verifiers behave and how they should be chosen in practice, and, in light of our clarifications, we kindly ask you to consider raising your score.

---

### Author Response · Authors · 2025-12-04

We thank all reviewers for their time, thoughtful feedback, and constructive suggestions. We are encouraged that reviewers highlighted our comprehensive and rigorous experimental design (ibxX, hm9P, VDYQ, Y1Lr), the clear motivation and empirical support for our verifier gain metric (ibxX, hm9P, Y1Lr), and confirmations of our hypotheses about solver-verifier behavior that can benefit future work (VDYQ, Y1Lr).

We provided detailed responses to each reviewer and revised the manuscript as follows.
- In Section 5.3, we expanded our explanation of Figure 6’s construction to improve interpretability and reproducibility.
- In Section 3.2, we clarified the motivation, assumptions, and usage of the **verifier gain** metric.
- In Section 5.3, we incorporated prior work on self-enhancement bias to further motivate our study.
- We added Appendix H to present correlation plots between verifier precision and the verifier’s own solver accuracy across different verification settings, clarifying why precision is not informative for understanding the lack of self-verification gains.
- To address questions about contributions, we revised Section 1 (Introduction) substantially to motivate the need for a broad study on verification beyond self-verification, situate our work within prior literature, highlight our central question in understanding the influence of a wide range of factors on verification effectiveness, and present our contributions as three concise, clearly explained points.

We also directly addressed reviewer Y1Lr’s thoughtful question regarding the practicality of verification. We clarified that verification is a well-established and effective method for improving solution quality, and that it offers advantages that neither solving nor ensembling alone can provide.

We thank the reviewers again for their valuable feedback and insightful questions.

---

### Meta-Review · Area_Chair_BJdQ · 2026-01-06

**Summary:**

This paper studies how performing rejection sampling with a verifier during inference helps (or does not help) performance of various LLMs. The study is comprehensive, spanning across 37 models and 9 benchmarks. The authors introduce a new metric, dubbed verifier gain, which can predict improvements from verifier-based rejection sampling.

Reviewers generally appreciated the comprehensiveness of the experiments (in terms of models/benchmarks), though some questions were raised with regard to why no closed-source frontier models were not evaluated. The experimental setup was also praised as thorough. The main weakness pointed out by all reviewers was that the study itself has limited novelty and the findings are not that surprising.

I agree with the reviewers---this is an empirically well-executed study but at times felt like just a bunch of numbers/plots without crisp motivation or takeaways. I think it would be more appropriate for a workshop than a full conference paper. (I also found the writing a bit hard to follow, in particular due to the authors deferring the definition of the various metrics to the appendix. However this was a minor issue and did not affect my decision that much.)

**Reviewer Concerns:**

Some concerns by reviewers included:
- limited novelty and unsurprising findings.
- lack of experiments on frontier closed models.
- experiments on benchmarks with limited practical value.
- lack of "so what" type takeaways.

The last point was somewhat addressed in the rebuttal (and in the updated version of the paper), but the remaining points were unaddressed.

**Reviewer Scores:**

I don't think any of the reviewers would have changed their scores.

---

### Decision · Program_Chairs · 2026-01-26

Reject